

# Mathematical model of voluntary vaccination against schistosomiasis

Santiago Lopez[1], Samiya Majid[1], Rida Syed[2,3], Jan Rychtar[2] and Dewey Taylor[2]

[1] Department of Biomedical Engineering, Virginia Commonwealth University, Richmond, VA, United States of America
[2] Department of Mathematics and Applied Mathematics, Virginia Commonwealth University, Richmond, VA, United States of America
[3] Department of Chemistry, Virginia Commonwealth University, Richmond, VA, United States of America

## ABSTRACT

Human schistosomiasis is a chronic and debilitating neglected tropical disease caused by parasitic worms of the genus Schistosoma. It is endemic in many countries in sub-Saharan Africa. Although there is currently no vaccine available, vaccines are in development. In this paper, we extend a simple compartmental model of schistosomiasis transmission by incorporating the vaccination option. Unlike previous models of schistosomiasis transmission that focus on control and treatment at the population level, our model focuses on incorporating human behavior and voluntary individual vaccination. We identify vaccination rates needed to achieve herd immunity as well as optimal voluntary vaccination rates. We demonstrate that the prevalence remains too high (higher than 1%) unless the vaccination costs are sufficiently low. Thus, we can conclude that voluntary vaccination (with or without mass drug administration) may not be sufficient to eliminate schistosomiasis as a public health concern. The cost of the vaccine (relative to the cost of schistosomiasis infection) is the most important factor determining whether voluntary vaccination can yield elimination of schistosomiasis. When the cost is low, the optimal voluntary vaccination rate is high enough that the prevalence of schistosomiasis declines under 1%. Once the vaccine becomes available for public use, it will be crucial to ensure that the individuals have as cheap an access to the vaccine as possible.

## INTRODUCTION

Human schistosomiasis is a chronic and debilitating neglected tropical disease caused by parasitic flatworms of the genus *Schistosoma* (*Ross et al., 2002*). It is endemic in many countries in Africa, South America, and Asia (*Madsen & Stauffe, 2022*). Worldwide there are an estimated 800 million people at risk of infection (*Steinmann et al., 2006*); over 230 million people are infected with about 201.5 million living in Africa (*Verjee, 2019*).

*Schistosoma* genus consists of 23 species (*Littlewood & Webster, 2017*); we will focus on *S. mansoni* which is endemic throughout sub-Saharan Africa. The life cycle of *Schistosoma mansoni* is described, for example in *McManus et al. (2018)*. The cycle involves an intermediate fresh-water snail host of *Biomphalaria* species (*Habib et al., 2021*) and the

Corresponding author
Jan Rychtar, rychtarj@vcu.edu

definitive human host. Eggs are excreted in the human faeces and they hatch upon contact with water. After hatching, the eggs release free-swimming ciliated larvae, miracidia which seek and penetrate snail hosts. Within the snails, the parasites develop into sporocysts which reproduce asexually to produce numerous larvae, called cercariae. The larvae emerge from snails in response to sunlight, and swim seeking human hosts. Once cercariae penetrate the skin of a human host their tails drop off and the larvae transform into schistosomula. They enter blood vessels and migrate to the liver, where they mature into adults. From the liver, the male and female worms migrate in pairs to the bowel. Females produce eggs which are excreted in faeces and the cycle continues.

Schistosomiasis control efforts include the following strategies:

1. disease treatment large-scale mass drug administration (MDA) of praziquantel (PZQ) (*Doenhoff et al., 2009*),
2. health education,
3. snail intermediate host control, and
4. water, sanitation and hygiene (WASH) programs (*Tchuenté et al., 2017*).

Successes in Japan, China, Egypt and in some sub-Saharan African countries such as Cameroon, Angola, Burkina Faso, Central African Republic, Chad, Congo, Mali, Senegal and Uganda demonstrate that control with progression towards elimination is possible (*Rollinson et al., 2013*). MDA by PZQ is a cost-effective 'preventive chemotherapy' and it is currently the strategy of choice and endorsed by WHO (*Tchuenté et al., 2017*; *WHO, 2021*). However, this strategy is unsustainable in the long term and interruptions in these MDA programs can lead to rebounds of egg count (*Ross et al., 2017*). Vaccines are being developed, but none are available yet (*Molehin, McManus & You, 2022*; *Molehin, 2020*; *Molehin et al., 2016*). The vaccine development faces many challenges, including the complexity of the schistosome life cycle, the parasite's ability to evade the immune system and the lack of adequate animal models for test trials (*Molehin, McManus & You, 2022*). Furthermore, there is a limited economic incentive to advance novel vaccine platforms as the disease affects the poorest regions of the world (*Molehin, McManus & You, 2022*).

Mathematical modeling plays a crucial and integral part of disease control and elimination (*Anderson & May, 1992*; *Behrend et al., 2020*). Many models exist for schistosomiasis transmission and control, including *Woolhouse, Hasibeder & Chandiwana (1996)*, *Spear et al. (2002)*, *Chiyaka & Garira (2009)*, *Zhou et al. (2013)*, *Mbah et al. (2014)*, *Stylianou et al. (2017)*, *Lo et al. (2018)*, *Gurarie et al. (2018)*, *Kadaleka, Abelman & Tchuenche (2021)*, *Kadaleka et al. (2021)*, *Kadaleka, Abelman & Tchuenche (2022)*, *Madubueze et al. (2022)* and *Ronoh et al. (2021)*. In *Collyer et al. (2019)* and *Kura et al. (2020)*, the authors modeled the impact of schistosomiasis vaccine. They found that in high transmission settings, MDA alone is unable to achieve the WHO goals of morbidity control and elimination as a public health problem. However, vaccination is able to achieve both goals in combination with MDA. Other models focus on snail intermediate hosts (*Woolhouse, 1991*; *Woolhouse & Chandiwana, 1990*; *Feng, Li & Milner, 2002*; *Allen & Victory Jr, 2003*; *Zhao & Milner, 2008*; *Mangal, Paterson & Fenton, 2008*; *Anderson, Loker & Wearing, 2021*). In *French et al. (2010)*, the authors fitted a model to data from a large-scale administration of PZQ in Uganda.

The aim of this paper is to focus on incorporating human behavior and voluntary individual vaccination against schistosomiasis. We want to determine whether voluntary vaccination alone could eliminate schistosomiasis as a public health concern, *i.e.,* decrease the prevalence of high intensity infections under 1% (*WHO, 2022*). We extend a compartmental model presented in *Gao et al. (2011)* which investigated the effect of MDA on schistosomiasis transmission. Inspired by *Stylianou et al. (2017)*, *Kura et al. (2019)*, we assume the vaccination is already available. We focus on what happens when MDA is no longer in place; similarly to modeling the post-MDA development in other NTDs such as trachoma (*Barazanji et al., 2023*), lymphatic filariasis (*Rychtář & Taylor, 2022*), or yaws (*Kimball et al., 2022*). Even if the vaccination is incorporated into existing pediatric vaccine programs and made mandatory by the government, it does not automatically mean that the population would adhere to the mandates. Vaccine hesitancy and avoidance is a real concern in the US (*Tolsma, 2015*), Europe (*Reczulska, Tomaszewska & Raciborski, 2022*) as well as Africa (*Anjorin et al., 2021*). For example, Central Africa has a significantly lower COVID-19 vaccine acceptance rate (less than 35%) than Southern Africa (about 75%) (*Anjorin et al., 2021*). There is a conflict between individual freedom and interests and the public health benefits (*Paplicki et al., 2018*). The vaccination, if adopted by enough people in the population, produces herd immunity and decreases the disease prevalence. This benefit can be enjoyed even by those not vaccinated (*Serpell & Green, 2006*). Thus, vaccination programs are prone to free-riding (*Ibuka et al., 2014*) because individuals maximize their self-interests (such as avoiding the costs associated with vaccination), rather than the interests of the entire group (*Maskin, 1999*).

We apply the game theory framework popularized in *Bauch & Earn (2004)*. The framework has been applied to many diseases; see *Wang et al. (2016)*, *Verelst, Willem & Beutels (2016)* and *Chang et al. (2020)* for recent reviews. As argued in *Wang et al. (2016)*, epidemics models incorporating human behavior provide more insight and better predictions. Thus, the game theory models have been applied to study the prevention and elimination of many NTDs, mpox (formerly monkeypox, *Bankuru et al. (2020)*, *Augsburger et al. (2022)*, *Augsburger et al. (2023)*, chikungunya (*Klein et al., 2020*), typhoid fever (*Acosta-Alonzo et al., 2020*), Chagas disease (*Han et al., 2020*), visceral leishmaniasis (*Fortunato et al., 2021*), lymphatic filariasis (*Rychtář & Taylor, 2022*), rabies (*Campo et al., 2022*), yellow fever (*Caasi et al., 2022*), or zika (*Angina et al., 2022*).

In the ideal case, the interests of the individual—to minimize one's costs, or to maximize one's benefits—align with the interest of the entire population—to reduce the prevalence of the disease below a certain threshold such as 1% for children age 5–14 (*WHO, 2022*). If this is the case, by behaving optimally (in their own sense), the individuals will behave optimally from the public health perspective. Thus, the individuals will more likely adhere to the mandatory vaccination policy and contribute to disease elimination as the public health concern. However, because interests can differ, a behavior that is optimal from the perspective of an individual may not be optimal from the perspective of the group and vice versa. To avoid confusion, in the rest of the paper, when we say "optimal", we will mean optimal from the individual perspective, unless specified otherwise.

## MATERIAL AND METHODS

We introduce a mathematical model for voluntary vaccination against schistosomiasis. First, we incorporate a possible vaccination into a compartmental model of schistosomiasis transmission developed by *Gao et al. (2011)*. Then, we add the game theory component that will allow us to investigate individuals' optimal vaccination decisions.

### Compartmental model

The human population is divided into susceptible ($S_1$), infectious ($I_1$) and vaccinated ($V_1$). The snail population is divided into susceptible ($S_2$) and infected ($I_2$). The schistosomiasis pathogen is divided into the snail-penetrating stage miracidia ($M$), and the human-penetrating stage, cercariae ($P$).

Human individuals are born susceptible to schistosomiasis at a rate $\Lambda_1$. Susceptible individuals become infected through contact with free-living cercariae present in contaminated fresh water. Because of saturating and crowding effect, we use a Holling type II incidence rate $\frac{\beta_1 P}{1+\alpha_1 P}$ (*Holling, 1959*; *Real, 1977*), where $\beta_1$ is the rate of transmission in small concentrations of $P$ and $\alpha_1$ is a scaling factor.

As in *Gao et al. (2011)*, we assume that the infected humans are treated at rate $\eta$, returning to the susceptible population; without treatment the individuals stay infected.

Susceptible individuals are vaccinated at a rate $\nu$. Vaccinated individuals are assumed immune against the disease. They lose their vaccine-induced immunity at a rate $\omega$ and become susceptible again. Infected humans may get vaccinated as well. From a practical standpoint, individuals with low intensity of infection will likely consider themselves susceptible and would vaccinate. Nevertheless, we assume that the vaccine does not work in these instances and the individuals stay infected. Infected humans release parasite eggs giving rise to the population of *miraccidia M* at rate $\gamma_1$; we ignore the egg hatching period.

Susceptible snails are born at rate $\Lambda_2$. They become infected at a rate $\frac{\beta_2 M S_2}{M_0 + \epsilon M^2}$ which is a Holling Type III incidence rate (*Holling, 1959*; *Real, 1977*), where $\beta_2$ is the rate transmission in small concentrations of $M$ and $M_0$ and $\epsilon$ are scaling factors. Infected snails give rise to the population of cercariae $P$ at a rate, $\gamma_2$.

For simplicity, we assume that the risk of contracting schistosomiasis after the age $\mu_1^{-1}$ is negligible. Thus, all humans are removed from the population at risk at a rate $\mu_1$ as they age. The infected cases also suffer from the disease-related death rate $\delta_1$; so they are removed from the population at a total rate $\mu_1 + \delta_1$. The susceptible snails die at a rate $\mu_2 + \theta$, where $\mu_2$ is the natural death rate and $\theta$ is the elimination rate of snails. Infected snails die at a rate $\mu_3 + \delta_2 + \theta$, where $\delta_2$ is the disease-related death rate of snails. miracidia ($M$) die at a rate $\mu_3$. The death rate of cercariae population $P$ is $\mu_4 + \tau$ where $\mu_4$ is the natural death rate and $\tau$ is the elimination rate. We ignore the negligible removal rates of miracidia and cercariae due to human and snail infections.

The transmission dynamic is illustrated in Fig. 1. The notation is summarized in Table 1.

The model yields the following differential equations.

$$\frac{dS_1}{dt} = \Lambda_1 - \frac{\beta_1 P S_1}{1+\alpha_1 P} - \mu_1 S_1 + \eta I_1 - \nu S_1 + \omega V_1, \qquad (1)$$

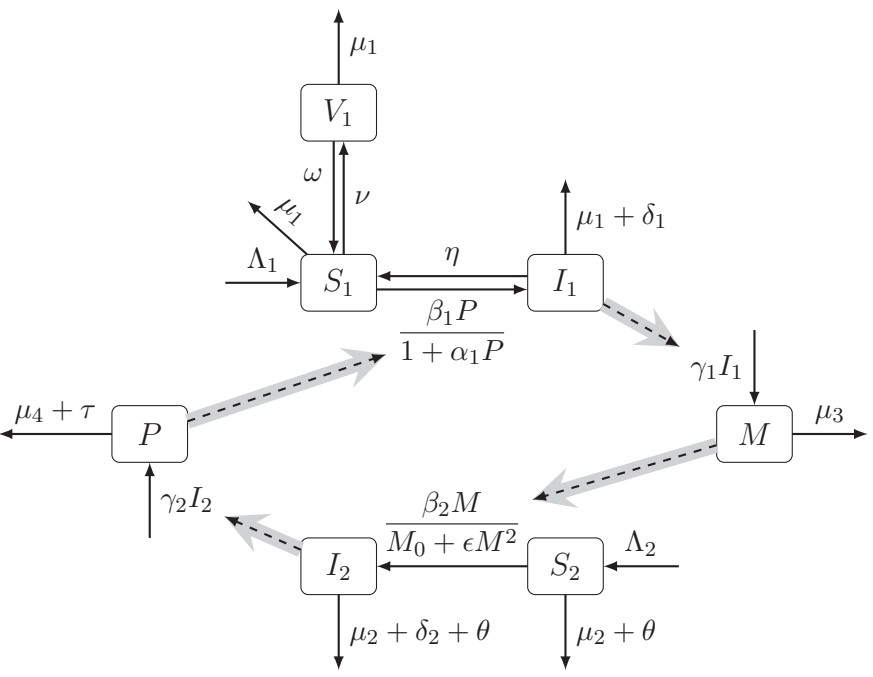

**Figure 1** **The diagram of the schistosomiasis transmission based on** *Gao et al. (2011)*. Human population is divided into Susceptible ($S_1$), Infected ($I_1$), and Vaccinated ($V_1$). Snail population is divided into Susceptible ($S_2$) and Infected ($I_2$). The free pathogens are divided into miracidia ($M$) and cercariae ($P$). The full arrows between compartments represent the transitions with the per capita rates. The dashed arrows show the influences of compartments on the transition rates. The gray arrow shows a schematic life cycle of the schistosomiasis pathogen..

$$\frac{dI_1}{dt} = \frac{\beta_1 P S_1}{1 + \alpha_1 P} - (\mu_1 + \delta_1 + \eta)I_1, \tag{2}$$

$$\frac{dM}{dt} = \gamma_1 I_1 - \mu_3 M, \tag{3}$$

$$\frac{dS_2}{dt} = \Lambda_2 - (\mu_2 + \theta)S_2, \tag{4}$$

$$\frac{dI_2}{dt} = \frac{\beta_2 M S_2}{M_0 + \epsilon M^2} - (\mu_2 + \delta_2 + \theta)I_2, \tag{5}$$

$$\frac{dP}{dt} = \gamma_2 I_2 - (\mu_4 + \tau)P, \tag{6}$$

$$\frac{dV_1}{dt} = \nu S_1 - \omega V_1 - \mu_1 V_1. \tag{7}$$
**Table 1  Model parameters (top part) and other notation (bottom part) as based on *Gao et al. (2011)*.** The rates are per capita per year, the times are in years. The calibration procedure is described in section "Model calibration".

| Symbol | Meaning | Value | Range | Source |
|---|---|---|---|---|
| $\Lambda_1$ | Birth rate (humans) | 0.031 | $[0.02, 0.04]$ | *World Bank (2022)* |
| $\mu_1^{-1}$ | Max age of people at risk | 20 | $[15, 25]$ | *Jordan (1972)* |
| $\mu_2$ | Natural death rate (snails) | 1.85 | $[1.5, 2.4]$ | *Appleton (1977)* |
| $\mu_3$ | Natural death rate (miracidia) | 1460 | $[1100, 1750]$ | *Maldonado, Acosta-Matienzo et al. (1948)* |
| $\mu_4$ | Natural death rate (cercariae) | 830 | $[500, 1100]$ | *Whitfield et al. (2003)* |
| $\gamma_1$ | Miracidia production rate | $1.1 \times 10^5$ | $[10^5, 2 \times 10^5]$ | *Alwan & LoVerde (2021)* |
| $\gamma_2$ | Cercariae production rate | $1.55 \times 10^5$ | $[0.9 \times 10^5, 2.2 \times 10^5]$ | *Gabrielli & Garba Djirmay (2022)* |
| $\delta_1$ | Disease related mortality rate (humans) | $10^{-4}$ | $[0, 10^{-2}]$ | *WHO (2021)* |
| $\eta$ | MDA treatment rate of humans | 0 | – | Assumed |
| $\tau$ | Elimination rate of cercariae | 0 | – | Assumed |
| $\theta$ | Elimination rate of snails | 0 | – | Assumed |
| $\nu$ | Vaccination rate | variable | $[0, 0.1]$ | Assumed |
| $\omega$ | Vaccine waning rate | $1/6.5$ | $[1/8, 1/5]$ | *Zhang et al. (2014)* |
| $\delta_2$ | Disease related mortality rate (snails) | 0.25 | $[0, 0.5]$ | Fitted |
| $\beta_1$ | Human infection rate by cercariae | 0.0013 | $[0.001, 0.0015]$ | Fitted |
| $\alpha_1$ | Scaling factor for human infection rate | 0.0315 | $[0.01, 0.05]$ | Fitted |
| $\beta_2$ | Snails infection rate by miracidia | 12.71 | $[10, 15]$ | Fitted |
| $M_0$ | Scaling factor for snail infection rate | 3500 | $[3000, 5000]$ | Fitted |
| $\varepsilon$ | Scaling factor for snail infection rate | 1.689 | $[1, 2]$ | Fitted |
| $\Lambda_2$ | Birth rate (snails) | 10 | $[5, 15]$ | Fitted |
| $c$ | Cost of vaccine relative to cost of schistosomiasis | 0.02 | $[0, 0.1]$ | Assumed |
| $d_1$ | Rate out of $I_1$ | $\mu_1 + \delta_1 + \eta$ | | |
| $d_2$ | Rate out of $S_2$ | $\mu_2 + \theta$ | | |
| $d_3$ | Rate out of $I_2$ | $\mu_2 + \delta_2 + \theta$ | | |
| $d_4$ | Rate out of $P$ | $\mu_4 + \theta$ | | |
| $\gamma$ | Auxiliary variable | $\frac{\Lambda_1 \gamma_1}{M_0}$ | | |
| $\delta$ | Auxiliary variable | $\gamma_2 \Lambda_2$ | | |
| $\alpha_2$ | Auxiliary variable | $\frac{\varepsilon}{M_0}$ | | |

## Game theory component

We add a game theory component to study individual vaccination based on the framework introduced in *Bauch & Earn (2004)* for childhood diseases and used in many other settings, including for the recent COVID-19 (*Choi & Shim, 2021*) or mpox outbreaks (*Augsburger et al., 2022*).

The game is played by susceptible individuals. As in *Bauch & Earn (2004)*, we assume the players are rational, act in their own best self-interests, and have complete information about the schistosomiasis epidemic. The individuals decide whether to vaccinate or stay unvaccinated. The payoff is a function of the individual's vaccination status and the vaccination status of the rest of the population. The payoff incorporates the cost of the vaccination (relative to the cost of the infection which can be assumed as 1 unit), $c$, the risk of getting infected, $\pi_{NV}$ if not vaccinated and $\pi_V$ if vaccinated. To evaluate the probability

of getting infected, we assume that the epidemics reached a steady state with $P^*$ cercariae given later by Eq. (59); $P^*$ depends on $v$, the vaccination rate in the population, but not on the decision of the focal individual. The probability that an unvaccinated individual becomes infected

$$\pi NV(v) = \frac{\frac{\beta_1 P*}{1+\alpha_1 P*}}{\frac{\beta_1 P*}{1+\alpha_1 P*} + \mu_1}, \tag{8}$$

where $\frac{\beta_1 P*}{1+\alpha_1 P*} + \mu_1$ is a rate at which susceptible individuals with no intention to vaccinate leave the Susceptible compartment and $\frac{\beta_1 P*}{1+\alpha_1 P*}$ is the rate at which they acquire the infection. Similarly, the probability of infection of a vaccinated individual is given by

$$\pi V(v) = \frac{\omega}{\omega + \mu_1} \pi_{NV}(v), \tag{9}$$

where the first term represents the probability of the vaccine waning during the individual's lifetime.

The solution of the game, called the Nash equilibrium, is the population-level vaccination rate—denoted $v_{NE}$—at which no individual can increase their own benefits by deviating from the population strategy. It follows that either (1) $v_{NE} = 0$ when $\pi_{NV}(0) \le \pi_V(0) + c$, *i.e.*, when the expected cost of not vaccinating is smaller than the expected cost of vaccinating in a population where nobody else vaccinates, or (2) $v_{NE}$ solves $\pi_{NV}(v) = \pi_V(v) + c$, *i.e.*, when the expected payoffs of not-vaccinating or vaccinating are equal. Here, $\pi_{NV}(0)$ is evaluated from Eq. (8) by substituting $v = 0$ for the vaccination rate and solving for the equilibria of the system Eqs. (10)–(16) as done in the later sections; the probability $\pi_V$ is evaluated analogously by Eq. (9); $c$ is the cost of vaccine relative to cost of schistosomiasis infection, *i.e.*, $C_{vaccine}/C_{Schistosomiasis}$. Thus, while not explicitly written, $P^*$ in Eq. (8) is a function of $v$ and thus $\pi_{NV}(v) = \pi_V(v) + c$ is an equation involving $v$ even after we substitute for $\pi_{NV}$ and $\pi_V$ from Eqs. (8) and (9).

## Analysis
### Simplification of the ODE
As in *Gao et al. (2011)*, we simplify the ODEs by introducing the following dimensionless variables: $S_1 = \Lambda_1 S_1, I_1 = \Lambda_1 I_1, V_1 = \Lambda_1 V_1, S_2 = \Lambda_2 S_2, I_2 = \Lambda_2 I_2, \mathbf{M} = M_0 M, \mathbf{P} = P$, $d_1 = \mu_1 + \delta_1 + \eta, d_2 = \mu_2 + \theta, d_3 = \mu_2 + \delta_2 + \theta, \gamma = \frac{\Lambda_1 \gamma_1}{M_0}, d_4 = \mu_4 + \tau, \delta = \gamma_2 \Lambda_2, \alpha_2 = \frac{\epsilon}{M_0}$. For simplicity, we discard the bold notation and we obtain the following system.

$$\frac{dS_1}{dt} = 1 - \frac{\beta_1 P S_1}{1+\alpha_1 P} - \mu_1 S_1 + \eta I_1 - v S_1 + \omega V_1, \tag{10}$$

$$\frac{dI_1}{dt} = \frac{\beta_1 P S_1}{1+\alpha_1 P} - d_1 I_1, \tag{11}$$

$$\frac{dV_1}{dt} = v S_1 - \omega V_1 - \mu_1 V_1, \tag{12}$$

$$\frac{dM}{dt} = \gamma I_1 - \mu_3 M, \tag{13}$$

$$\frac{dS_2}{dt} = 1 - \frac{\beta_2 M S_2}{1 + \alpha_2 M^2} - d_2 S_2, \tag{14}$$

$$\frac{dI_2}{dt} = \frac{\beta_2 M S_2}{1 + \alpha_2 M^2} - d_3 I_2, \tag{15}$$

$$\frac{dP}{dt} = \delta I_2 - d_4 P. \tag{16}$$

### Disease-free equilibrium

The equilibria of the dynamics Eqs. (10)–(16) are obtained by setting the time derivatives to 0 and solving the following system of algebraic equations.

$$0 = 1 - \frac{\beta_1 P S_1}{1 + \alpha_1 P} - \mu_1 S_1 + \eta I_1 - \nu S_1 + \omega V_1, \tag{17}$$

$$0 = \frac{\beta_1 P S_1}{1 + \alpha_1 P} - d_1 I_1, \tag{18}$$

$$0 = \nu S_1 - \omega V_1 - \mu_1 V_1, \tag{19}$$

$$0 = \gamma I_1 - \mu_3 M, \tag{20}$$

$$0 = 1 - \frac{\beta_2 M S_2}{1 + \alpha_2 M^2} - d_2 S_2, \tag{21}$$

$$0 = \frac{\beta_2 M S_2}{1 + \alpha_2 M^2} - d_3 I_2, \tag{22}$$

$$0 = \delta I_2 - d_4 P. \tag{23}$$

There are two equilibria of the dynamics: the disease-free equilibrium and the endemic equilibrium.

Setting $I_1 = I_2 = M = P = 0$, the system Eqs. (17)–(23) reduces to

$$0 = 1 - \mu_1 S_1 - \nu S_1 + \omega V_1, \tag{24}$$

$$0 = 1 - d_2 S_2, \tag{25}$$

$$0 = v S_1 - \omega V_1 - \mu_1 V_1. \tag{26}$$

By Eq. (25), $S_2^0 = \frac{1}{d_2}$. Adding Eq. (24) and Eq. (26) gives $1 = \mu_1(S_1 + V_1)$. Thus, the disease-free equilibrium is given by

$$S_1^0 = \frac{\mu_1 + \omega}{\mu_1(\mu_1 + v + \omega)}, \tag{27}$$

$$S_2^0 = \frac{1}{d_2}, \tag{28}$$

$$V_1^0 = \frac{v}{\mu_1(\mu_1 + v + \omega)}. \tag{29}$$

### *Effective reproduction number*

The effective reproduction number, $\mathcal{R}$, is found by the next generation matrix method (*van den Driessche & Watmough, 2002*).

There are four compartments carrying infections, $I_1, M, I_2, P$ and we will keep them in this order. The rate of new infections is given by

$$\mathcal{F} = \left[ \frac{\beta_1 P S_1}{1 + \alpha_1 P}, 0, \frac{\beta_2 M S_2}{1 + \alpha_2 M^2}, 0 \right]^T. \tag{30}$$

Differentiating $\mathcal{F}$ at the disease-free equilibrium, we obtain

$$F = \begin{bmatrix} 0 & 0 & 0 & \beta_1 S_1^0 \\ 0 & 0 & 0 & 0 \\ 0 & \beta_2 S_2^0 & 0 & 0 \\ 0 & 0 & 0 & 0 \end{bmatrix}. \tag{31}$$

The other transmissions in the system are given by

$$\mathcal{V} = \left[ -d_1 I_1, \gamma I_1 - \mu_3 M, -d_3 I_2, \delta I_2 - d_4 P \right]^T. \tag{32}$$

Differentiating $\mathcal{V}$ at the disease-free equilibrium, we obtain

$$V = \begin{bmatrix} -d_1 & 0 & 0 & 0 \\ \gamma & -\mu_3 & 0 & 0 \\ 0 & 0 & -d_3 & 0 \\ 0 & 0 & \delta & -d_4 \end{bmatrix}. \tag{33}$$

Thus,

$$V^{-1} = \begin{bmatrix} -\dfrac{1}{d_1} & 0 & 0 & 0 \\ -\dfrac{\gamma}{d_1\mu_3} & -\dfrac{1}{\mu_3} & 0 & 0 \\ 0 & 0 & -\dfrac{1}{d_3} & 0 \\ 0 & 0 & -\dfrac{\delta}{d_3 d_4} & -\dfrac{1}{d_4} \end{bmatrix}, \tag{34}$$

and

$$FV^{-1} = \begin{bmatrix} 0 & 0 & -\dfrac{S_1^0\beta_1\delta}{d_3 d_4} & -\dfrac{S_1^0\beta_1}{d_4} \\ 0 & 0 & 0 & 0 \\ -\dfrac{S_2^0\beta_2\gamma}{d_1\mu_3} & -\dfrac{S_2^0\beta_2}{\mu_3} & 0 & 0 \\ 0 & 0 & 0 & 0 \end{bmatrix}. \tag{35}$$

The largest eigenvalue of $FV^{-1}$ is

$$\mathcal{R} = \rho(FV^{-1}) = \sqrt{\dfrac{S_1^0 S_2^0 \beta_1 \beta_2 \delta \gamma}{d_1 d_3 d_4 \mu_3}} \tag{36}$$

$$= \sqrt{\dfrac{(\mu_1+\omega)\beta_1\beta_2\delta\gamma}{d_1 d_2 d_3 d_4 \mu_1 \mu_3 (\mu_1+\nu+\omega)}}. \tag{37}$$

The disease-free equilibrium is locally asymptotically stable if $\mathcal{R} < 1$ and the endemic equilibrium is stable if $\mathcal{R} > 1$ (*van den Driessche & Watmough, 2002*).

## Critical vaccination rates

The value of $\nu$ needed to eliminate schistosomiasis can be found by solving

$$\mathcal{R} = \sqrt{\dfrac{(\mu_1+\omega)\beta_1\beta_2\delta\gamma}{d_1 d_2 d_3 d_4 \mu_1 \mu_3 (\mu_1+\nu+\omega)}} < 1 \tag{38}$$

for $\nu$. It follows that whenever $\nu > \nu_{HI}$, where

$$\nu_{HI} = \max\left\{0, (\mu_1+\omega)\left(\dfrac{\beta_1\beta_2\delta\gamma}{d_1 d_2 d_3 d_4 \mu_1 \mu_3} - 1\right)\right\}, \tag{39}$$

then $\mathcal{R} < 1$ and the disease can be eliminated.

### Endemic equilibrium

Here we find solutions of the system Eqs. (17)–(23) for the endemic equilibrium with the pathogen still present in the environment. By Eq. (19),

$$V_1 = \dfrac{\nu S_1}{\omega + \mu_1}. \tag{40}$$

Adding Eqs. (17)–(19) yields

$$1 = \mu_1(S_1 + V_1) - (d_1 - \eta)I_1. \tag{41}$$

Thus,

$$S_1 = \frac{1 - (d_1 - \eta)I_1}{\mu_1\left(1 + \frac{\nu}{\omega + \mu_1}\right)} = (1 - (d_1 - \eta)I_1)S_1^0. \tag{42}$$

By Eq. (20),

$$M = \frac{\gamma I_1}{\mu_3}. \tag{43}$$

By Eq. (21),

$$1 + \alpha_2 M^2 = \beta_2 M S_2 + d_2(1 + \alpha_2 M^2)S_2. \tag{44}$$

Thus,

$$\frac{S_2}{1 + \alpha_2 M^2} = \frac{1}{\beta_2 M + d_2(1 + \alpha_2 M^2)} = \frac{\mu_3^2}{\beta_2 \gamma \mu_3 I_1 + d_2(\mu_3^2 + \alpha_2 \gamma^2 I_1^2)}. \tag{45}$$

By Eq. (22),

$$I_2 = \frac{\beta_2 M}{d_3} \frac{S_2}{1 + \alpha_2 M^2} = \frac{\beta_2 \gamma \mu_3 I_1}{d_3[\beta_2 \gamma \mu_3 I_1 + d_2(\mu_3^2 + \alpha_2 \gamma^2 I_1^2)]}. \tag{46}$$

By Eq. (23),

$$P = \frac{\delta I_2}{d_4} = \frac{\delta \beta_2 \gamma \mu_3 I_1}{d_3 d_4[\beta_2 \gamma \mu_3 I_1 + d_2(\mu_3^2 + \alpha_2 \gamma^2 I_1^2)]}. \tag{47}$$

Plugging Eqs. (42) and (47) into (18), or, equivalently, into $d_1 I_1 + \alpha_1 P d_1 I_1 = \beta_1 P S_1$, and then simplifying, yields the following cubic equation for $I_1$

$$I_1^*(a_1 I_1^{*2} + a_2 I_1^* + a_3) = 0, \tag{48}$$

where

$$a_1 = \frac{d_1 d_2 d_3 d_4 \alpha_2 \gamma^2}{S_1^0}, \tag{49}$$

$$a_2 = \beta_2 \gamma \mu_3 \left(\delta \beta_1(d_1 - \eta) + \frac{d_1 d_3 d_4}{S_1^0} + \frac{d_1 \alpha_1 \delta}{S_1^0}\right), \tag{50}$$

$$a_3 = \frac{d_1 d_2 d_3 d_4 \mu_3^2}{S_1^0} - \delta \beta_1 \beta_2 \gamma \mu_3. \tag{51}$$

Note that

$$a_3 = \beta_1 \beta_2 \delta \gamma \mu_3 \left(\frac{1}{\mathcal{R}^2} - 1\right). \tag{52}$$

Thus, $a_3 < 0$ if and only if $\mathcal{R} > 1$. Consequently, Eq. (48) has a unique positive root

$$I_1^* = \frac{-a_2 + \sqrt{a_2^2 - 4a_1 a_3}}{2a_1} \tag{53}$$

if and only if $\mathcal{R} > 1$. Once $I_1^*$ is given by Eq. (53) as a solution of Eq. (48), the other compartments are given by

$$S_1^* = (1 - (d_1 - \eta)I_1^*)S_1^0, \tag{54}$$

$$V_1^* = \frac{\nu S_1^*}{\omega + \mu_1}, \tag{55}$$

$$M^* = \frac{\gamma I_1^*}{\mu_3}, \tag{56}$$

$$S_2^* = \frac{1 + \alpha_2 M^{*2}}{\beta_2 M^* + d_2(1 + \alpha_2 M^{*2})}, \tag{57}$$

$$I_2^* = \frac{\beta_2 M^*}{d_3} \frac{S_2^*}{1 + \alpha_2 M^{*2}}, \tag{58}$$

$$P^* = \frac{\delta I_2^*}{d_4}. \tag{59}$$

### Finding optimal individual vaccination strategy

To find a Nash equilibrium, we have to solve

$$\pi_{NV}(\nu) = \pi_V(\nu) + c \tag{60}$$

where $\pi_{NV}$ and $\pi_V$ are given by Eqs. (8) and (9), respectively. Rearranging Eq. (60) yields

$$\frac{\frac{\beta_1 P^*}{1 + \alpha_1 P^*}}{\frac{\beta_1 P^*}{1 + \alpha_1 P^*} + \mu_1} = \frac{c}{1 - \frac{\omega}{\omega + \mu_1}}. \tag{61}$$

We solve it for $P^*$ to get

$$P^* = \frac{c\mu_1}{\beta_1 - \frac{\beta_1 \omega}{\omega + \mu_1} - c\beta_1 - c\mu_1\alpha_1}. \tag{62}$$

Since $P^*$ is given by Eq. (59), we get

$$I_2^* = \frac{d_4 c\mu_1}{\delta(\beta_1 - \frac{\beta_1 \omega}{\omega + \mu_1} - c\beta_1 - c\mu_1\alpha_1)}. \tag{63}$$

From now on, we will use previous calculations to express $v$ in terms of $I_2^*$ given by Eq. (63). By Eqs. (56), (57), and (58), $I_1^*$ can be expressed in terms of $I_2$ as

$$I_1^* = \frac{(1-d_3I_2^*)\beta_2 \pm \sqrt{\left(\beta_2(1-d_3I_2^*)\right)^2 - 4\left(\frac{I_2^* d_2 d_3 \alpha_2 \gamma}{\mu_3}\right)\left(\frac{I_2^* d_2 d_3 \mu_3}{\gamma}\right)}}{2\frac{I_2^* d_2 d_3 \alpha_2 \gamma}{\mu_3}}. \tag{64}$$

By Eq. (48), we can express $S_1^0$ in terms of $I_1^*$ as

$$S_1^0 = \frac{d_1 d_2 d_3 d_4 \alpha_2 \gamma^2 I_1^{*3} + d_1 \beta_2 \gamma \mu_3 (d_3 d_4 + \alpha_1 \delta) I_1^{*2} + d_1 d_2 d_3 d_4 \mu_3^2 I_1^*}{\beta_1 \beta_2 \delta \gamma \mu_3 (I_1^* - (d_1 - \eta) I_1^{*2})}, \tag{65}$$

and, by Eq. (27), we can express $v$ in terms of $S_1^0$ as

$$v = \frac{\mu_1 + \omega - S_1^0 \mu_1^2 - S_1^0 \mu_1 \omega}{S_1^0 \mu_1}. \tag{66}$$

Hence, the Nash equilibrium is given by Eq. (66), where $S_1^0$ is given in Eq. (65), $I_1^*$ is given by Eq. (64), and $I_2^*$ is given by Eq. (58).

## Model calibration

We focus on transmission of *S. mansoni* and we locate as many parameters specific to this species as possible. However, since *S. haematobium* is also endemic in sub-Saharan Africa, some parameter estimates are based on that species or simply schistosoma species in general; we specifically say so if it is the case. We perform sensitivity and uncertainty analysis to account for possible discrepancies in parameter values.

For birth rate, we will use a country in sub-Saharan Africa, like Zimbabwe where schistosomiasis in general is endemic (*Midzi et al., 2014*). In Zimbabwe, the birth rate is 31 births per 1,000 people per year (*World Bank, 2022*), *i.e.*, $\Lambda_1 = 0.031$.

The egg output of cases infected by *S. haematobium* (*Bradley & McCullough, 1973*) as well as the length of water contact (*Jordan, 1972*) varies by age and there is a sharp drop off after the age 20 for both measures (*Kura et al., 2021*). We will thus assume the same is true for *S. mansoni* and consider the aging rate $\mu_1 = 1/20$.

We will consider snails of the *Planorbidae* family, especially *Biomphalaria* species, as they are one are a common intermediate host of schistosomiasis (*Gabrielli & Garba Djirmay, 2022*). Their life span ranges between 5 to 8 months (*Appleton, 1977*) and we use the average death rate $\mu_2 = 12/6.5 \approx 1.85$ per year. The longevity of *S. mansoni* miracidia is relatively small, about 5-6 h and no more than 9 h (*Maldonado, Acosta-Matienzo et al., 1948*). We will thus use $\mu_3 = 365/(6/24) = 1,460$. Similarly, *S. mansoni* cercariae live on average about 10.5 h with a range from 8-17 h (*Whitfield et al., 2003*) and so we set ($\mu_4 = 365 \times 24/10.5 \approx 830$). We note that the cercariae may survive up to 72 h (*Nelwan, 2019*).

*S. mansoni* females release about 300 eggs per day (*Alwan & LoVerde, 2021*; *Mooee, Sandgeound et al., 1956*); we will thus use $\gamma_1 = 300 * 365 \approx 1.1 \times 10^5$.

The number of *S. mansoni* cercariae produced daily is 250–600 (*Gabrielli & Garba Djirmay, 2022*). We will thus use $\gamma_2 = 425 * 365 \approx 1.55 \times 10^5$.

We estimate the disease related mortality as $\delta_1 = 1/10^4$ based on 2016 global schistosomiasis data of 24,000 deaths and 240 million infections (*Gabrielli & Garba Djirmay, 2022*; *WHO, 2021*). This is in general agreement with *Kheir et al. (1999)* who estimated the annual mortality between $50/10^5$ and $1/1000$ (or higher for specific kinds of infections.

There is currently no vaccine (*Molehin, McManus & You, 2022*) for humans. Nevertheless, based on phase 1 clinical trials in baboons, the longevity of one of the tested vaccines is 5–8 years (*Zhang et al., 2014*). We thus set the vaccine waning rate to be $\omega = 1/6.5$. The vaccine reduces the parasitic female load by about 90%, but for simplicity we will assume a complete protection.

For the purpose of the model, we will assume $\eta = 0$ because PZQ helps to control morbidity by killing adult schistosomes but it is ineffective against juvenile worms (*McManus et al., 2018*; *Hagan et al., 2004*). We also assume $c = 0.02$ with the range $[0, 0.05]$, the cost of the vaccine is about 1/50 of the cost of contracting schistosomiasis (and somewhere between 0 and 1/20 of the cost of the disease).

To find the values of other parameters, we set the controls to 0, *i.e.*, set $\nu = 0, \theta = 0, \tau = 0, \eta = 0$, and fitted the model predictions to observed data of (a) the reproduction number, $\mathcal{R}_0 \approx \sqrt[4]{4.31}$ based on *Woolhouse, Hasibeder & Chandiwana (1996)*, (b) the proportion of infected individuals, $I_1/(I_1 + S_1) \approx 0.227$ based on *Midzi et al. (2014)*, and (c) the proportion of infected snails $I_2/(I_2 + S_2) \approx 0.018$ based on *Odongo-Aginya et al. (2008)*. We used MATLAB's built-in optimization procedure fmincon which is a nonlinear programming solver that returns a minimizer of a given function subject to various constraints. We note that *Woolhouse, Hasibeder & Chandiwana (1996)* estimated the average number of female worms that reach reproductive age produced by a typical female worm over the course of its life by 4.31. Our model has four stages of a parasite (miracidia, parasites in snails, cercaria, and parasites in humans). Thus, the $\mathcal{R}_0$ derived by the next-generation matrix method should satisfy $\mathcal{R}_0^4 \approx 4.31$ so that a typical miracidia produces on average 4.31 miracidia during the full cycle.

## RESULTS

For the parameter values specified in Table 1, the vaccination rate leading to elimination of schistosomiasis is given by $\nu_{HI} \approx 0.23$. This is illustrated in Fig. 2. It means that the entire population needs to be vaccinated in about 4.5 years, *i.e.*, slightly more frequently than the minimal vaccination waning rate.

The optimal voluntary vaccination rate is $\nu_{NE} \approx 0.16$. At this rate, the entire population would be vaccinated in about 6.25 years, just under the assumed waning rate.

The prevalence when individuals use the optimal voluntary vaccination is about 4.7%. We can thus see that after the termination of MDA and other control measures, schistosomiasis would not be eliminated as a public health concern (currently defined as < 1% proportion of heavy intensity schistosomiasis infections in school age children *WHO (2021)*) by optimal voluntary vaccination alone.

Figure 3A shows the dependence of the optimal voluntary vaccination rate $\nu_{NE}$ on the relative cost of vaccination, $c$. Once the cost of vaccination grows above about 0.053,

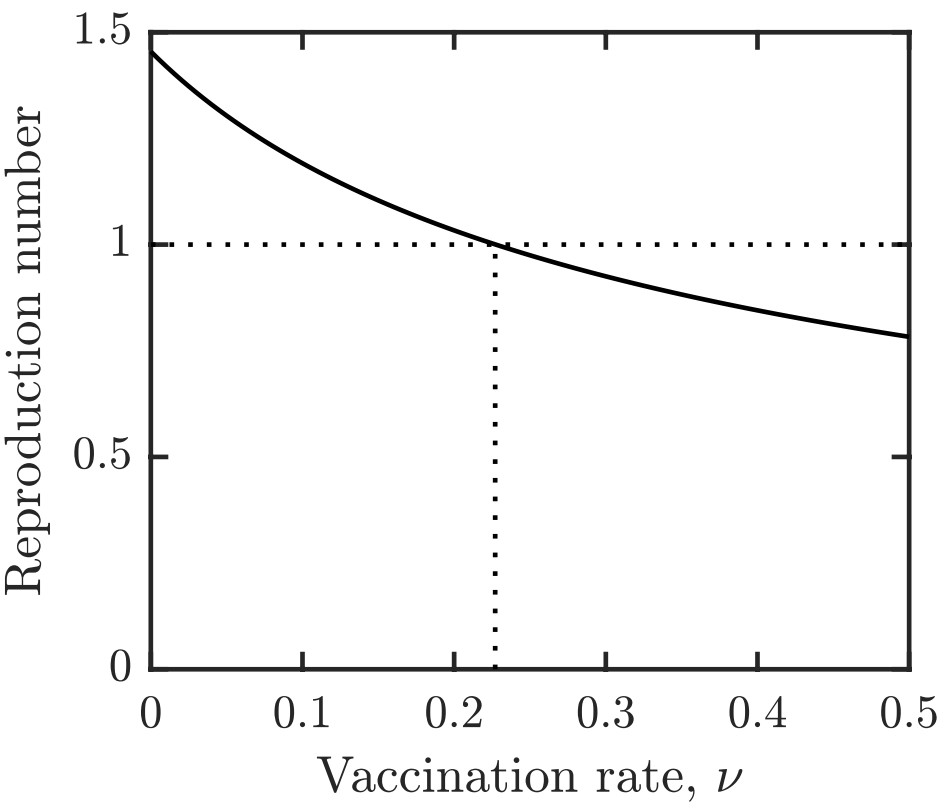

**Figure 2** **Dependence of reproduction number $\mathcal{R}$ on the vaccination rate, $v$.** Other parameters as in Table 1.

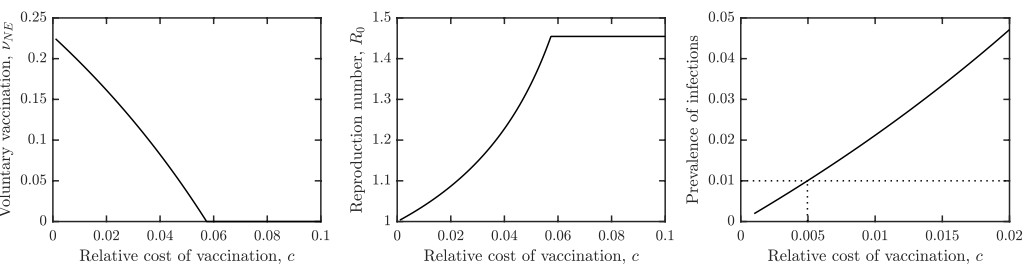

**Figure 3** **The dependence of (A) the optimal voluntary vaccination rate, $v_{NE}$, (B) the effective reproduction number $\mathcal{R}$ at the optimal voluntary vaccination rate, and (C) the prevalence of infections at the optimal voluntary vaccination rate, on the relative cost of vaccination, $c$.** Other parameters as in Table 1.

$v_{NE} = 0$. It means that if the cost of vaccination is higher than about $1/20$ of the cost of schistosomiasis infection, it is not beneficial for the individuals to vaccinate. On the other hand, when the cost of vaccination is very low, then $v_{NE} \approx v_{HI}$, meaning that schistosomiasis would be very close to elimination.

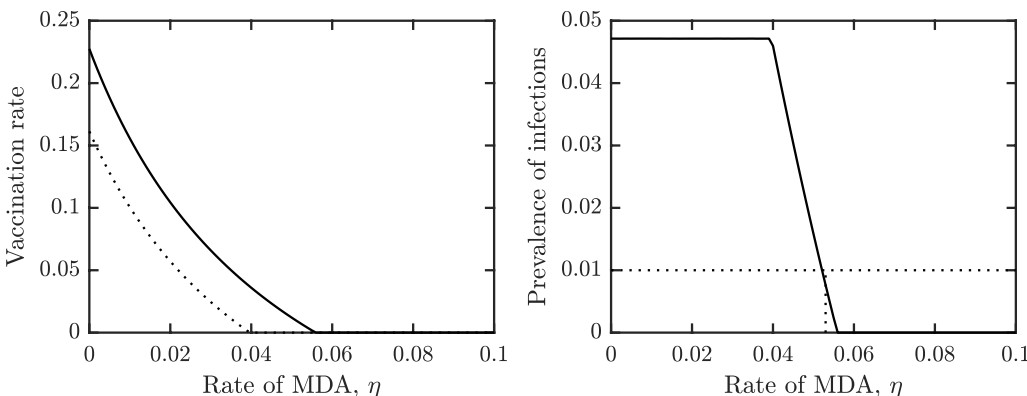

**Figure 4** The dependence of (A) the optimal voluntary vaccination rate, $v_{NE}$ (dotted) and the vaccination rate needed for her immunity, $v_{HI}$ (full line), (B) the prevalence of infections at the optimal voluntary vaccination rate, on the MDA rate, $\eta$. Other parameters as in Table 1.

Similarly, Fig. 3B shows the dependence of the effective reproduction number on $c$. In agreement with Fig. 3A, when $c \approx 0$, $\mathcal{R} \approx 1$ and when $c > 0.053$, $\mathcal{R} \approx 1.45$. Note that as long as $c > 0$, $\mathcal{R} > 1$, *i.e.*, the optimal voluntary vaccination will never completely eliminate schistosomiasis on its own. Finally, Fig. 3C shows the dependence of the prevalence on $c$. It follows that as long as $c < 0.005$, the prevalence is less than 1%, *i.e.*, schistosomiasis would be considered eliminated as a public health concern.

Figure 4 shows how the outcomes depend on the MDA rate, $\eta$. The optimal voluntary vaccination rate, $v_{NE}$ is positive for $\eta < 0.04$ while the vaccination rate needed for her immunity, $v_{HI}$ is positive for $\eta < 0.057$. Moreover, the prevalence of schistosomiasis infections when everybody adopts the optimal voluntary vaccination rate is constant (and higher than 4%) for $\eta < 0.04$. It follows that when using a combination of MDA and vaccination, the schistosomiasis can be eliminated, but in most cases it would be eliminated by MDA alone.

## Uncertainty and sensitivity analysis

We used the Latin hyper-cube sampling with partial rank correlation coefficient(LHS-PRCC) scheme (*Blower & Dowlatabadi, 1994*; *Saltelli et al., 2004*) to complete the uncertainty and sensitivity analysis. A full description of this method can be found in *Marino et al. (2008)*.

Figure 5 shows the results of the analysis for $\mathcal{R}$, $v_{HI}$ and $v_{NE}$. The uncertainty shows the distribution of model prediction among all the sampled parameter values. The most frequent values of $\mathcal{R}$ are between 0.6 and 2 with a peak around 1.2; we note that these are for vaccination rates between 0 and 0.1. The most frequent values of $v_{HI}$ are between 0 and 0.5 with most vaccination rates being below 0.25. Taken together, we can see that schistosomiasis would most likely be eliminated as long as the vaccination rates are 0.25 per year or higher, *i.e.*, one would need to vaccinate the entire population at risk within 4 years. On the other hand, the optimal voluntary vaccination rate peaks between 0 and 0.03

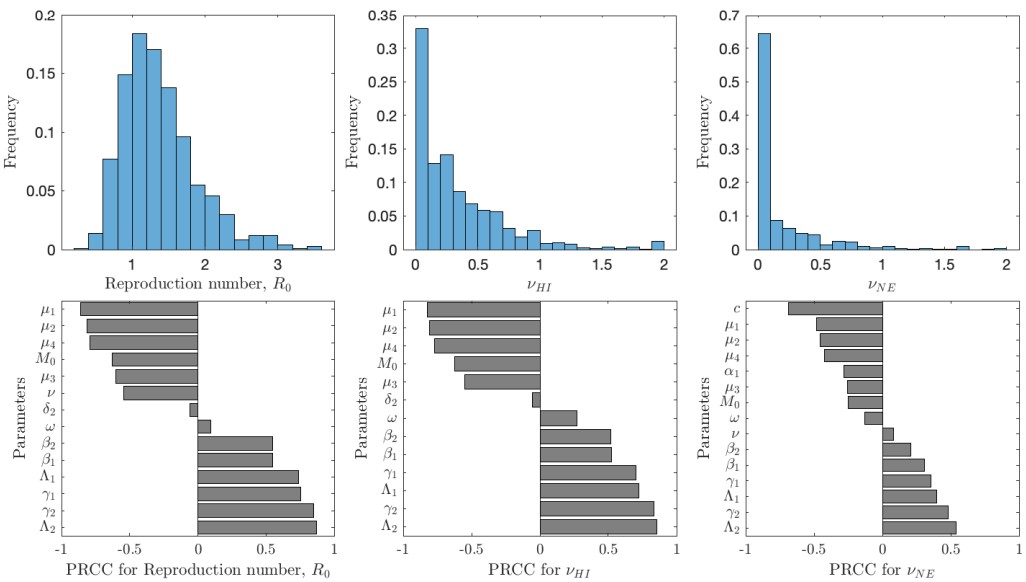

**Figure 5** **The uncertainty (top row) and sensitivity (bottom row) analysis of the reproduction number $\mathcal{R}$ (left), vaccination rate needed for herd immunity $\nu_{HI}$ (center) and the optimal voluntary vaccination rate $\nu_{NE}$ (right).** Parameter ranges are as in Table 1.

and most values are less than 0.1. There is thus a big difference between the vaccination rate needed to eliminate schistosomiasis and the rate that is optimal for the individual.

The reproduction number is most sensitive to the aging rate $\mu_1$, the death rates of snails, $\mu_2$, cercariae $\mu_4$ and miracidia $\mu_3$, vaccination rate $\nu$, and the scaling factor for snail infection rate $M_0$; an increase of any of these parameters would cause decrease of $\mathcal{R}$. Similarly, increase of snail birth rate $\Lambda_2$, cercariae production $\gamma_2$, miracidia production $\gamma_1$ or human birth rate $\Lambda_1$ would cause $\mathcal{R}$ to decrease. Finally, an increase of $\beta_1$ or $\beta_2$, *i.e.,* increase of the rates parasites attack humans or snails, causes $\mathcal{R}$ to increase. The sensitivity analysis of $\nu_{HI}$ and $\nu_{NE}$ follows a very similar pattern. The exception is that the voluntary vaccination rate is most sensitive to the cost $c$, the higher the cost, the lower the vaccination rate.

The value of $\nu_{NE}$ is not as important as the actual prevalence of schistosomiasis when everybody adopts the optimal voluntary strategy. As seen from Fig. 6, in about 35% of the cases, the prevalence is below 1%; however, in about 65% of the cases, the prevalence is higher than 1%, meaning that schistosomiasis would not be eliminated as a public health concern. The prevalence is most sensitive to the cost of the vaccination (relative to the cost of schistosomiasis).

The situation improves significantly when the vaccination is accompanied by other control measures as seen in Fig. 7. In about 75% of the cases, the prevalence is below 1%. The prevalence is most sensitive to the rate of MDA treatment. The dependence on the other controls (elimination of snails or cercariae) is negligible.

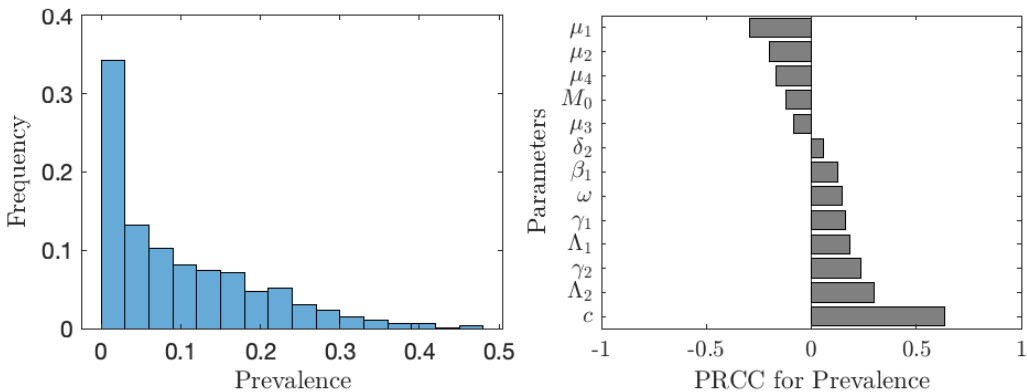

**Figure 6  The uncertainty (left) and sensitivity (right) analysis of the prevalence of schistosomiasis when everybody uses the optimal voluntary vaccination rate.** Parameter ranges are as in Table 1.

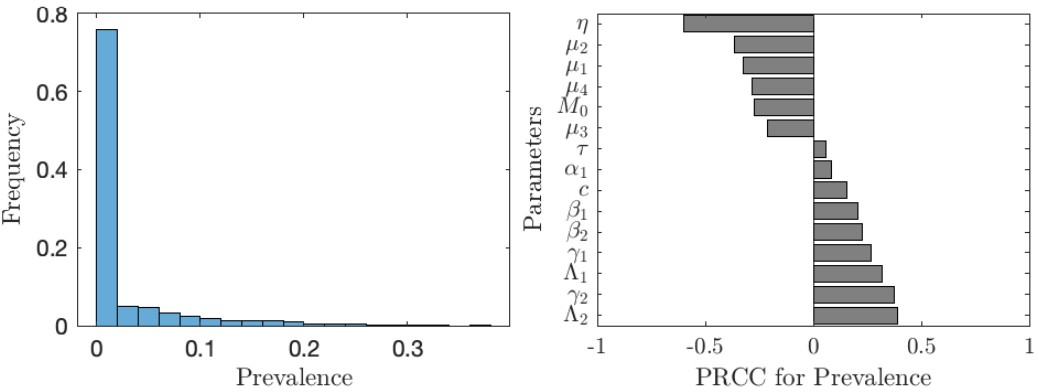

**Figure 7  The uncertainty (left) and sensitivity (right) analysis of the prevalence of schistosomiasis when everybody uses the optimal voluntary vaccination rate.** Parameter ranges are as in Table 1; however, this time the control measures (MDA treatment rate of humans, $\eta$, elimination rate of cercariae, $\tau$, elimination rate of snails, $\theta$) vary between 0 and 0.1 per year.

## DISCUSSION

The big caveat of our quantitative results, though, is that, for simplicity, our model did not incorporate several important feature of schistosomiasis. First, the age is an important factor influencing the water contact and infection rates (*Kura et al., 2021*), but we considered it only marginally. To incorporate the age-dependent water contact properly, we would have to stratify the human population by age groups. This stratification would also allow better tracking of the prevalence of the infections amongst school age children, which is crucial for the WHO's elimination goal. The age groups play an important role from the logistical standpoint as well. Like MDA which is administered mostly to school age children (*King et al., 2011*), the vaccines would have to be administered before age 5 by incorporating into existing pediatric vaccine programs. Due to waning protection, the vaccination would have to be administered every 5 or so years. However, these aspects were not addressed by our

model and thus more modeling effort need to be done to properly understand the effects of the vaccine.

Second, we assumed the vaccine offers 100% protection while the real efficacy will be likely around 90% (*Zhang et al., 2018*). Nevertheless, based on modeling of imperfect vaccine done for example in *Reluga & Galvani (2011)*, *Augsburger et al. (2023)*, *Augsburger et al. (2022)*, as long as the vaccine is 85% or more effective, there are no big differences in model outcomes between perfect and imperfect vaccines. Furthermore, usage of *S. mansoni*-only vaccine would likely not be acceptable in sub-Saharan Africa as there are regions where both *S. mansoni* and *S. haematobium* are endemic. A model that accounts for both species at the same time would be needed to better understand what to do in those regions.

Third, individuals eventually reach immunity (*Kura et al., 2021*; *Wilkins et al., 1984*) and this was omitted in our model that concentrated on the young population only. While the recovered compartment should be added to the later iterations of the model, we believe its addition would not significantly alter the results.

Our model can be further improved in several other ways. The underlying compartmental model can be made more realistic by (a) adding "exposed" compartments to human and intermediate hosts (such as in *Anderson, Loker & Wearing, 2021*), (b) considering the fact that infected humans release eggs rather than miracidia, and most importantly(c) specifically model the parasite load (such as in *Woolhouse, Hasibeder & Chandiwana (1996)*). Also, schistosomiasis endemicity exhibits a great variation when even neighboring villages show vastly different levels of parasite loads (*Carabin et al., 2005*). The distribution of schistosoma infections are highly over dispersed among hosts, even within age groups (*Bundy, 1988*); this can have implications on how effective the vaccination program is in reality. Incorporating some sort of structural modeling network to epidemics, for example as done in *Hadjichrysanthou & Sharkey (2015)* would be helpful. The game theory part of the model can be extended as follows. We assumed that every individual has the same risk of infection. However, the risk varies by age and by their behavioral pattern (*MBra et al., 2018*). Individuals thus have different risk perceptions (*Poletti, Ajelli & Merler, 2011*) and also base their decision on different social aspects (*Xia & Liu, 2013*). Therefore, it is often beneficial to use multi-agent-simulation (MAS) methodology (*Iwamura & Tanimoto, 2018*; *Kabir & Tanimoto, 2019*; *Kuga, Tanimoto & Jusup, 2019*; *Kabir & Tanimoto, 2020*) which allows more flexibility and realism. Furthermore, our model assumed the risk of contracting the disease to be the only cost associated with not-vaccination. If the vaccine is made mandatory, there can also be penalties for vaccine avoidance, possibly shrinking the gap between optimal voluntary vaccination level and the level required to achieve elimination. Finally, we assumed all individuals have perfect and full information. This is unlikely to happen in reality. However, the people will look up to their local leadership for advice and support. It is thus critical for the success of the vaccination campaign that the local leaders receive proper information about the disease and the available prevention methods.

## CONCLUSIONS

We extended the compartmental model of schistosomiasis transmission (*Gao et al., 2011*) by adding the possibility of vaccination (*Molehin, McManus & You, 2022*; *Stylianou et al., 2017*) and applied the game-theoretic framework (*Bauch & Earn, 2004*). Unlike previous models of schistosomiasis transmission that focused on control and treatment at the population level, our model focuses on incorporating human behavior and voluntary individual vaccination.

We identified vaccination rates needed to achieve the herd immunity as well as optimal (from the individuals' perspective) voluntary vaccination rates. We evaluated the prevalence of schistosomiasis for the scenario when everyone uses the optimal vaccination rates. We demonstrated that the prevalence remains too high (higher than 1%) unless the vaccination costs are sufficiently low. Thus, we can conclude that the voluntary vaccination alone may not be sufficient to eliminate schistosomiasis as a public health concern. When combining vaccination with MDA, the elimination is feasible; however, in such scenarios, the elimination would be possible by MDA alone.

We calibrated our model based on the data from literature. However, especially data related to transmission rates were lacking and we thus had to fit our model numerically to empirical data. We argue that there is an ongoing need to strengthen data collection and evaluation for decision-making (*Toor et al., 2021*). We also performed uncertainty and sensitivity analysis and showed that the results are relatively robust; the optimal voluntary vaccination (without MDA) will not eliminate schistosomiasis in at least 65% of the scenarios. With MDA, the situation is somewhat better, the elimination would occur in all but 25% of the scenarios.

The cost of the vaccine for the individual was an important factor determining whether or not voluntary vaccination can yield the elimination of schistosomiasis. When the cost is low (*e.g.*, subsidized by governments or international help), the optimal voluntary vaccination rate is high enough that the prevalence of schistosomiasis declines under 1% and the disease is thus eliminated as a public health concern. Once the vaccine becomes available for public use, it will therefore be crucial to ensure that the individuals have cheap access to the vaccine.

Our main finding that voluntary vaccination alone may not be enough to eliminate schistosomiasis is not surprising. These conclusions had been already reached in a general scenario (*Geoffard & Philipson, 1997*) as well as demonstrated for specific diseases with a high cost of vaccination relative to the cost of the disease such as cholera (*Kobe et al., 2018*), Hepatitis B (*Chouhan et al., 2020*; *Scheckelhoff et al., 2021*), lymphatic filariasis (*Rychtář & Taylor, 2022*), polio (*Cheng et al., 2020*), or typhoid fever (*Acosta-Alonzo et al., 2020*).

### Funding

The authors received no funding for this work.

## Competing Interests

The authors declare there are no competing interests.

## Author Contributions

- Santiago Lopez conceived and designed the experiments, performed the experiments, analyzed the data, prepared figures and/or tables, authored or reviewed drafts of the article, and approved the final draft.
- Samiya Majid conceived and designed the experiments, performed the experiments, analyzed the data, prepared figures and/or tables, authored or reviewed drafts of the article, and approved the final draft.
- Rida Syed conceived and designed the experiments, performed the experiments, analyzed the data, prepared figures and/or tables, authored or reviewed drafts of the article, and approved the final draft.
- Jan Rychtar conceived and designed the experiments, performed the experiments, analyzed the data, prepared figures and/or tables, authored or reviewed drafts of the article, and approved the final draft.
- Dewey Taylor conceived and designed the experiments, performed the experiments, analyzed the data, prepared figures and/or tables, authored or reviewed drafts of the article, and approved the final draft.

## Data Availability

The MATLAB code used to generate the figures is available in the Supplemental File.

## Supplemental Information

Supplemental information for this article can be found online at http://dx.doi.org/10.7717/peerj.16869#supplemental-information.

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
