# Peer review of "Mathematical model of voluntary vaccination against schistosomiasis"

_PeerJ, doi:10.7717/peerj.16869_

## Round 0.1 · original submission · Minor Revisions

The reviewers have written several detailed comments on each section of the manuscript to facilitate amendments that will improve the manuscript. The reviewers appear to be in agreement that the manuscript is deserving of publication but only following the suggested modifications.

**Language Note:** The review process has identified that the English language must be improved. PeerJ can provide language editing services - please contact us at copyediting@peerj.com for pricing (be sure to provide your manuscript number and title). Alternatively, you should make your own arrangements to improve the language quality and provide details in your response letter. – PeerJ Staff

·

Basic reporting

no comment

Experimental design

The model formulation is okay. The work has many punctuation marks and errors, especially the equations. The equations are part of the statement.

Some terms are not defined in the work.
For instance, what is V_{NE} after (9)?
How do the authors get \pi_{NV} (0)?
What is the meaning of V_{NE}=0? How do V_{NE} solves \pi_{NV} (v)=\pi_{v} +c?

What is c?

Are the numbers (1) and (2) on Page 10 the same as the ones in the model formulation?

Validity of the findings

no comment

·

Basic reporting

The manuscript addressed a complex mathematical model. Some minor adjustments are needed for the narrative parts and use of English with grammar and sentece formation.

The references seemed appropriate and relative.

The professional article structure was quite good, but long strings of symbols needed to be clearer. I suggest a list where many terms/symbols are used.

The authors clearly considered possibilities but did provide an argument, and included limitations.

Experimental design

While the mathematical model was adapted from earlier published authors, the statistical formal were appropriate. They took me time to verify - two whole days!

While I would not say that the investigation was "rigorous", it was sufficiently plausible and credible. It would take an expert to explain the formulae used to a less discerning reader.

The methods were reproducible, perhaps with a different interpretation in other circumstances.

Validity of the findings

The impact has yet to be noted. The theory model (after Nash's Equilibrium Theory) is provided.

The underlying data was satisfactory.

The conclusions were satisfactory considering the limitations of the research.

Additional comments

A well thought out research project which would add to the literature as an idea, built on previous work.

Innovative model with possibilities, but much has to be worked out with the delivery and distribution of vaccines, control points, supplies, and other logistics.

The Nash Equilibrium Theory is well established and can be applied in this case, but it needs an explanation for the reader.

---

## Round 0.2 · accepted · Accept

Thank you for addressing all of the comments left by the two previous reviewers. They are satisfied that you have responded to all of their comments, and the manuscript is now ready to proceed for publication.

·

Basic reporting

The authors have addressed the comments I made.

Experimental design

The experimental design is okay.

Validity of the findings

The manuscript validate their findings in their explanation and conclusion.

·

Basic reporting

All the rebuttal discussion has been reviewed and deemed acceptable. Standards required have been met. Apart from tinkering with one or two phrases, the manuscript is now clearer and communicates well with amendments included by revision. The English used is now clearer and unambiguous.

One reference was deleted.

The article was professional.

MS was self-contained within the hypothesis.

Experimental design

This was an extension of previous literature with an original aspect.

The research question, while theoretical, bears thought and is meaningful.

The authors have attempted to advance a model of schistosomiasis infection decrease, moving towards elimination.

Methodology has already been approved.

Validity of the findings

Confidence in validity of the model.

Sufficient date provided to support theory

Conclusions are satisfactory

Additional comments

Good paper which will add to the literature.